# Machine Learning for Aiding Blood Flow Velocity Estimation Based on Angiography

**DOI:** 10.3390/bioengineering9110622

**Published:** 2022-10-28

**Authors:** Swati Padhee, Mark Johnson, Hang Yi, Tanvi Banerjee, Zifeng Yang

**Affiliations:** 1Department of Computer Science and Engineering, Wright State University, Dayton, OH 45435, USA; 2Department of Mechanical and Materials Engineering, Wright State University, Dayton, OH 45435, USA

**Keywords:** machine learning (ML), convolutional neural networks (CNN), computational fluid dynamics (CFD), optical flow method (OFM), dye perfusion, hemodynamics, least absolute shrinkage and selection operator (LASSO), angiography, cardiovascular, particle image velocimetry (PIV)

## Abstract

Computational fluid dynamics (CFD) is widely employed to predict hemodynamic characteristics in arterial models, while not friendly to clinical applications due to the complexity of numerical simulations. Alternatively, this work proposed a framework to estimate hemodynamics in vessels based on angiography images using machine learning (ML) algorithms. First, the iodine contrast perfusion in blood was mimicked by a flow of dye diffusing into water in the experimentally validated CFD modeling. The generated projective images from simulations imitated the counterpart of light passing through the flow field as an analogy of X-ray imaging. Thus, the CFD simulation provides both the ground truth velocity field and projective images of dye flow patterns. The rough velocity field was estimated using the optical flow method (OFM) based on 53 projective images. ML training with least absolute shrinkage, selection operator and convolutional neural network was conducted with CFD velocity data as the ground truth and OFM velocity estimation as the input. The performance of each model was evaluated based on mean absolute error and mean squared error, where all models achieved or surpassed the criteria of 3 × 10^−3^ and 5 × 10^−7^ m/s, respectively, with a standard deviation less than 1 × 10^−6^ m/s. Finally, the interpretable regression and ML models were validated with over 613 image sets. The validation results showed that the employed ML model significantly reduced the error rate from 53.5% to 2.5% on average for the v-velocity estimation in comparison with CFD. The ML framework provided an alternative pathway to support clinical diagnosis by predicting hemodynamic information with high efficiency and accuracy.

## 1. Introduction

According to the statistical data from the World Health Organization, cardiovascular associated diseases have become the topmost cause of death [1,2], and it has been identified that hemodynamic factors play a significant role in the development and progression of such diseases [3,4,5,6]. For example, the development of atherosclerosis is based on the observation that vascular inflammation and plaques are distributed near side branches or arterials stenosis, where blood flow is non-uniform, and at the lesser curvature of bends where the blood flow rate is relatively low [7,8]. The effect of blood flow on the vascular wall is through shear-stress forces, which influences the behavior of endothelial cells, including morphological adaptations and physiological changes.

The methods for estimation of wall shear stress (WSS) can be performed by acquiring three-dimensional (3D) reconstruction of vessel volume using either invasive modalities such as intravascular ultrasound and invasive coronary angiography, or less-invasive techniques, including computed tomography angiography and cardiac magnetic resonance angiography with the application of numerical methods to calculate flow within the reconstructed arterial model [9,10]. 3D reconstruction of the patient-specific vascular models with subsequent computational fluid dynamics (CFD) simulations can be used to predict the nature of blood flow in the human vascular system [11,12,13,14], with specific mesh generation and boundary conditions (BCs). The isolation and generation of BCs are among the most significant challenges in the integration of CFD for assessing the physiologic significance of artery disease [15]. However, the reconstruction of the vascular model using the medical images, the accuracy of the reconstructed geometry, and further the computation resources required for computation are not friendly to the wide application of this methodology in clinical community.

On the other hand, in vivo quantifying hemodynamics by the diagnostic imaging process is also widely investigated [16,17,18,19,20]. To determine hemodynamic characteristics, a high spatial resolution of the blood flow velocity distribution is needed, especially for quantifying the WSS. Premedical images based on digital subtraction angiography (DSA) can be used to estimate the blood flow velocity distribution using intensity-based algorithms, such as the optical flow method (OFM) [16,17,18,19,20]. Specifically, the movement of the contrast pattern is calculated from the DSA images, representing the movement of the local blood flow. However, such a method suffers low accuracy in the predictions of flow characteristics [21]. DSA images involve X-rays that transmit through the patient and produce a contrast projection onto the scintillator plane. The intensity variation is based on how much X-ray energy passes through the nonuniform medium, which results from varying attenuation through muscle, bone, soft tissue, blood, and contrast agent [22,23]. DSA images are generally acquired by transmitting light through a 3D domain, and then projected onto a 2D image. Without isolating a 2D plane within the flow, the images are essentially integrated representations of the entire flow field, which can introduce errors into the calculations, especially when the measurements are inferred as measurements at the center plane of the domain [10]. Kawaji et al. [24] calculated blood velocity by determining displacement over consecutive images with a known elapsed time interval between frames. The uncertainty in the velocity profiles begin to increase when the endpoint becomes unclear, but for simple flows the relative error can be as low as 2.5% [24]. In our earlier work, Yang and Johnson [21] investigated dye visualization specific to a pulsatile flow through a horizontal tube, where OFM underestimated the center plane velocity by 16~24% in the central region and by about 29~43% in the outer region compared to PIV. To increase the accuracy of calculating the velocity distribution, this research proposes the use of machine learning to correct velocity measurements from OFM. Alternatively, inspired by recent developments in data-driven scientific computing, big data, convolutional neural networks (CNN) and deep hidden physics algorithms [25,26], machine learning (ML) has been used in biomedical and bioengineering applications [27,28,29,30,31]. For instance, Raissi et al. have developed a physics-informed deep learning method which used passive scalar contours as input and encoded the Navier–Stokes equations into their algorithm [25]. There is growing potential of artificial intelligence (AI) in detecting hemodynamic information to aid clinical diagnosis by integrating preclinical images and CFD modeling [32,33].

To extend the use of ML, this study used optical flow velocity based on projective images to provide input for the ML model and therefore requires the physics-based flow equations to be solved once (at beginning) as opposed to every time an epoch is conducted. The ML model was used to correct high resolution flow details obtained from OFM to provide more accurate measurements in hemodynamics estimation. More specifically, this work employed CFD modeling to simulate dye perfusion through water, and the results were used to generate a projection-based image similar to DSA. The projective images and velocity distributions served as the input and the ground truth data, respectively, for ML training. The pixel intensity was determined through a discretized form of the light attenuation equation named Beer-Lambert Law [34]. Although blood is non-Newtonian (which was assumed as Newtonian and replaced by water as a pilot study), the purpose of this work was to test the efficiency and accuracy of ML algorithms velocity predictions based on projective images. The detailed framework of this study was shown in Figure 1, specifically in three steps, e.g., (1) simulate dye perfusion in water and generate projective images to provide the flow velocity estimation using OFM and the ground truth velocity from CFD, in which CFD modeling was validated by particle image velocimetry(PIV) experiments; (2) train ML models using the OFM and CFD data; and (3) validate the ML models using both CFD data and parallel in vitro experiments. The presented work has the potential to be employed into clinical applications to automate the flow velocity estimation with high efficiency and accuracy.

## 2. Materials and Methods

### 2.1. CFD and OFM Data

Choosing CFD to simulate the dye perfusion in water over experimental dye projection images can facilitate accurate velocity fields for the training of ML algorithms and quantify the error analysis between OFM and ML predictions. In addition, the simulation images can avoid random errors induced in the experiments. In this study, Cradle SC/Tetra 2021 (Software Cradle Co., Ltd., Osaka, Japan) was used to simulate the dye injection into a laminar water flow to obtain the accurate 3D velocity field and obviate experimental uncertainties related to pixel intensities. The accurate velocity field from CFD served as the ground truth for the supervised training of the ML model. The simulation was conducted within a 1 × 1-inch square cross-sectional tube with a length of 1-inch with the dye inlet located in the center (see Figure 2). To minimize computational costs, a fourth of the domain was employed into the simulation. In this work, the blood was simplified as water in simulations and experiments. The governing equations can be written in tensor form as:(1)$$\frac{\partial u_{i}}{\partial x_{i}}=0$$
(2)$$\rho \frac{\partial u_{i}}{\partial t}+\rho \frac{\partial \left(u_{j}u_{i}\right)}{\partial x_{j}}=-\frac{\partial p}{\partial x_{i}}+\mu \frac{\partial }{\partial x_{j}}\left[\frac{\partial u_{i}}{\partial x_{j}}+\frac{\partial u_{j}}{\partial x_{i}}\right]+\rho g_{i}$$
(3)$$\rho \frac{\partial \phi }{\partial t}+\rho \frac{\partial u_{i}\phi }{\partial x_{i}}=\rho D\frac{\partial }{\partial x_{i}}\left[\frac{\partial \phi }{\partial x_{i}}\right]$$
where $u_{i}$ represents the water/dye velocity, ρ is the density of water/dye which was identical to the measurements in the experiments, p is pressure, μ is viscosity of water/dye which was identical to the properties in the experiments, $g_{i}$ is gravity, ϕ is the dye concentration which is identified as a scalar value 1.0 in the dye injection inlet, D is the dye diffusion coefficient. The coefficient of diffusion was set at 10^−9^ m^2^/s as the typical value for dye between 10^−10^ and 10^−9^ m^2^/s [35]. A set of unstructured tetrahedral meshes was generated using the assembled meshing package in Cradle SC/Tetra 2021. A mesh independent study for the quarter model was conducted, and the final mesh with elements of 16,850,005 was adopted for the current study. The mesh size near the wall and in the dye injection region was refined with y+ < 1, where y+ is the dimensionless wall distance.

Four different transient simulations were conducted where the dye concentration and patterns varied in each simulation. Specifically, the first simulation injected dye at a constant rate of 0.005 m/s with a dye radius of 0.37 mm initially and gradually increased the radius throughout time until the radius reached 2.0 mm, which is the same situation as the parallel experiment. Second simulation had intermittent dye injections where dye was injected for 0.1 secs then stopped for 0.25 s, and this process would repeat continuously with a period of 0.35 s. The diameter of the dye was held constant at 2.0 mm for the second simulation. For the third simulation, the dye was injected intermittently in the same pattern as the second simulation, except the dye concentration was no longer uniform at the inlet. The distribution of the dye concentration was represented with a cosine function where the dye concentration at the center was 1.0, and near a radius of 2.0 mm the concentration was 0.54. The fourth simulation involved a uniform dye concentration in the radial direction but varied in a sinusoidal pattern with time. The dye distribution equations are expressed for the third and fourth simulation, respectively, as
(4)$$\phi \left(x\right)=\phi_{o}\cos\left(\omega x\right)$$
(5)$$\phi \left(t\right)=\cos\left(\frac{4\pi }{3}t\right)$$

In Equation (4), ϕ represents the dye concentration, $\phi_{o}$ represents a constant concentration value of $\phi_{o}$= 1.00, ω is the angular frequency of ω = 500/m. Equation (5) provides the transient cosine wave for the dye concentration, where $\frac{4\pi }{3}$ provides a period of 1.5 s and t is the time in seconds. These four variations were carried out to represent the impact of different perfusion patterns of the dye on the estimated velocity accuracy. Results from the simulation were exported either every 0.2 s or 0.3 s (in reference to the simulation time), and additional results were exported 0.02 s after the 0.2 or 0.3 s interval. This short time step was necessary for OFM to calculate the velocity between two successive images, as the nature of the OFM algorithm requires the movement of the pattern to be less than 1 pixel to maintain a decent accuracy. Overall, the four cases provided 106 exported result files which would lead to 106 image files or 53 data files after being analyzed with OFM.

Light passing through the domain was simulated to generate projective images. This was accomplished with the Beer-Lambert law [34] which describes the amount of light attenuated through a medium. By solving for the light intensity, *I*, out of the medium, the overall projection image can be created by dividing the domain into multiple cells and performing successive calculations of the intensity. Variable *z* represents the direction parallel to the light beam and perpendicular to the flow direction, *μ* represents the attenuation coefficient, and $I_{o}$ is the initial intensity prior to passing through the element,
(6)$$\frac{dI}{dz}=-\mu I$$
(7)$$I=I_{o}\exp\left(-\mu △z\right)$$

For example, if a single row of elements aligned in the z-direction encounters an initial light source, the first element will decrease the intensity based on the element’s attenuation coefficient (which is directly related to the dye concentration). The output intensity through the first element will then be used as the initial intensity for the second element. This process is continued until the light ray passes through all the elements. Therefore, the intensity can be written in a discretized form as shown below.
(8)$$I_{j}=I_{j-1}\exp\left(-\mu_{j}△z_{j}\right)$$

Example of a projective image is shown below in Figure 3, where the darker regions represent locations where dye has attenuated light. The image resolution is 161 × 306 pixels.

Once projective images were generated, OFM was applied to each image pair to calculate velocity vectors at each pixel. The governing equation for OFM is an expression of a brightness constraint, which assumes the intensities are constant from one image to the next.
(9)$$Q_{t}+V\cdot \nabla Q+Q\nabla \cdot V=D\nabla^{2}Q$$

Here, the intensity of pixel is represented by *Q* (a matrix for the entire image), $Q_{t}$ is the temporal brightness gradient, the velocity vector is given by ***V***, and *D* is the diffusion coefficient. Additional details can be found in our previous paper.

Along with the u- and v-velocity components, the intensity gradients in the x- and y-directions were also calculated. Since OFM calculates movements based on pixel intensities, the gradients are of importance as it provides an insight into which vectors were calculated based on low or high intensity gradients. The velocity calculated from OFM is regarded as the test data while the CFD velocities are regarded as the ground truth data for the ML model. The CFD velocities were extracted from the center plane of the simulation as the goal is to improve the OFM results to better represent the velocities in the middle plane of the domain. Once these files were created, they were then used as the training data for the ML algorithm. 423,093 simulated data points representing the flow velocity were generated for the current modeling.

### 2.2. Experimental Data for Validation

The data discussed above is generated based on CFD results, however, the future application of this work is to apply this method directly to in vivo images such as DSA. Direct application to DSA images is not feasible currently because no method can accurately describe the flow field for validations. Therefore, an in vitro experiment analogous to DSA was created to obtain real images for testing the current method against accurate PIV measurements. Additionally, this provides insight into whether the ML algorithm can correct velocity values that incur error from the in vitro experiment such as non-uniform light intensity throughout the image domain. The experiment consists of a square vertical tube test section (with the same dimension as the counterpart used in the CFD simulation) with a water reservoir and contraction region directly above, as presented in Figure 4. This contraction region leads water from the reservoir to the test section through a gradual cross-sectional change to mitigate the inducing of turbulence. Downstream of the test section is a flow control valve that controls the speed of the flow when coupled with a constant water height in the water reservoir. A pump feeds water into the reservoir where a partition wall, which is slightly shorter than the reservoir walls, allows water to spill over into the adjacent reservoir to maintain a constant water level. Dye is injected in the center of the square tube through the guidance of a centering device to maintain consistent positioning during the injection process. Finally, a convex lens is positioned one focal length away from the LED bulb, where the lens’ position is between the bulb and the test section, so light rays can transmit parallel through the test section thus mitigating non-uniform light intensity on the image.

In the experiments, the water is analogous to blood in DSA images and the dye provides contrast similar to the contrast agent injected in blood vessels. Furthermore, the LED light is comparable to the X-ray source since both transmit through the domain and project contrast agent perfusion onto a 2D plane. A LaVision PIV system consisted of a Nd: Yag laser with a wavelength of 532 nm with a maximum pulse energy of 300 mJ, and a programmable timing unit (PTU), and a 14-bit CCD camera with a resolution of 1392 × 1040 pixel (66 × 49 mm) was employed to conduct PIV measurements. The laser was shaped into a laser sheet of 1 mm in the measurement region to illuminate the flow field. The PTU synchronized the laser and the camera to obtain the particle images with a time delay of 33 ms between images. The flow was seeded with fluorescent red polyethylene microspheres with a diameter ranging from 10 to 45 μm. The fluorescent emission light from the particle has a wavelength of 610 nm. A long pass filter with a cut-off wavelength of 600 nm was mounted on the camera length to filter out the laser light and only leave the emission light passing through. This fluorescent imaging can effectively remove the light reflection noise from the wall. DaVis 8.0 was used to process the images to obtain 1000 instantaneous velocity distributions with a frame rate of 30 Hz. A multi-pass cross correlation analysis was conducted in DaVis 8.0 with a 32 × 32 pixels interrogation region with 50% overlap, then a second pass with an 8 × 8 pixels window size with 50% overlap. The final velocity vector resolution is 5.2 vectors/mm. The uncertainty in the velocity measurements is estimated to be about 1% of the magnitude. A total of 5071 flow velocity points for OFM velocity and PIV velocity were generated for evaluation of current ML algorithms.

The CFD simulation of the velocity distribution is compared with the PIV measurements in the middle cut plane, as presented in Figure 5. Three lines were extracted from PIV and CFD: 5 mm, 15 mm, and 35 mm. The overall difference in the velocity distribution between the CFD simulation and the PIV measurement is less than 5% of the velocity magnitude, which essentially validated the accuracy of the current simulation results (see Table 1).

### 2.3. Machine Learning Model

The following sections detail predicting the accurate velocity field with supervised neural network training. Three deep machine learning models with varying numbers of parameters and depths were tested. Regression is a predictive modeling task that involves predicting a numerical output given some input. It is different from classification tasks that involve predicting a class label. Typically, a regression task involves predicting a single numeric value. Although, some tasks require predicting more than one numeric value. These tasks are referred to as multiple-output regression or multi-output regression for short. In multi-output regression, two or more outputs are required for each input sample, and the outputs are required simultaneously. The assumption is that the outputs are a function of the inputs, and u- and v-components of the velocity are correlated. Since OFM calculates a velocity vector for every point in the image, even when the dye is not present (pseudo vectors were calculated in regions without dye), specific regions with the presence of dye were selected for the training and test data. The images generated were 8-bit grayscale images where pixel intensity values are assigned between 0 and 255. The presence of dye in a pixel location decreases the intensity value for that location. Based on observation, pixel values of 80 or less (after considering the image noise) represent regions where dye is dominant and thus represent data points used in the training data or test data sets. This criterion was applied to all the OFM output results, and each valid data point compiles into a single file. Multiple data points correspond with the same XY-coordinates as a result of multiple time instances containing dye at that point. Therefore, all of the duplicates except one are used for training, and the other duplicate was used in the test set. For instance, if the coordinate point (x, y) = (0.001, 0.0253) is near the dye inlet, then there were multiple OFM data sets that contain dye at that coordinate point. Assuming 10 duplicates, one data point was selected randomly for the test set while the remaining 9 points were compiled into the training data. This random selection provided unbiased data for the test set to determine if the ML algorithm was able to predict values accurately.

#### 2.3.1. LASSO Regression Model

A multivariate (multi-input, multi-output) regression model with the L1 penalty was designed, commonly known as LASSO [36]. We selected this model since it achieves sparsity in the estimated model by setting the regression coefficients for features to zero for those features that do not affect the output or target values. We treat this ML model as our baseline model.

#### 2.3.2. Multi-Layer Perceptron Model

A multilayer perceptron (MLP) model is defined for the multi-output regression task defined in the previous section. Each sample has 4 inputs and 2 outputs; therefore, the network requires an input layer that expects 4 inputs in the first hidden layer and 2 nodes in the output layer. The ReLU activation function was used in the hidden layer which has 32 nodes, chosen after some trial and error. The model was fitted using two loss functions and the Adam version of stochastic gradient descent.

#### 2.3.3. CNNs for Multi-Output Regression

CNNs have proven great capability of learning important features from images at the pixel level in order to make useful predictions for both classification and regression problems [37]. Another advantage of this approach, compared to conventional fully connected layer networks, lies in the fact that convolutions provide weight-sharing and sparse connectivity. These properties enable more efficient memory usage to learn the necessary information needed to create a surrogate model. The surrogate model then reconstructs an approximation of the entire velocity field from a given set of boundary conditions. Standard CNN has several convolutional layers followed by fully connected layers (layers where each hidden node is connected to every hidden node in the preceding and subsequent layer), ending with a classification softmax layer (that generates the final classification output). Adapting a classification CNN architecture to regression consists of removing the softmax layer and replacing it with a fully connected regression layer with linear or sigmoid activation. Linear activation means that the transfer function is a straight line. Thus, the activation is proportional to the input and not confined to a specific range. Three convolutional layers were added with ReLU activation and a max pool layer after the first convolutional layer.

#### 2.3.4. Long Short-Term Memory (LSTM) Model

LSTM is a type of Recurrent Neural Network (RNN), i.e., a multi-layer NN. The LSTM architecture was originally introduced by Hochreiter and Schmidhuber [38] with the purpose of overcoming the vanishing or exploding gradients problem. In a network of *n* hidden layers, *n* derivatives are multiplied together. If the derivatives are large then the gradient will increase exponentially as we propagate down the model until they eventually explode, known as the problem of exploding gradient. Alternatively, if the derivatives are small then the gradient will decrease exponentially as we propagate through the model until it eventually vanishes, known as the vanishing gradient problem. LSTM allows flow gates, i.e., the input gate, the forget gate, the control gate, and the output gate, as shown in Figure 6. The input gate, the forget gate, the control gate, and the output gate are denoted by $i_{t}$, $f_{t}$, $c_{t}$, and $o_{t}$, respectively. The details of these four gates are enlightened below.

The input gate, forget gate and control gate are expressed as:(10)$$i_{t}=\sigma \left(w_{i}\times \left[h_{t-1},x_{t}\right]+b_{i}\right)$$
(11)$$f_{t}=\sigma \left(w_{f}\times \left[h_{t-1},x_{t}\right]+b_{f}\right)$$
(12)$$\tilde{c_{t}}=\tanh\left(w_{c}\times \left[h_{t-1},x_{t}\right]+b_{c}\right)$$
(13)$$c_{t}=f_{t}\times c_{t-1}+i_{t}\times \tilde{c_{t}}$$

The input gate decides which information can be transferred from the earlier cell to the current cell as shown in Equation (10). The forget gate is defined by Equation (11), and it is used to store the information from the input of previous memory or otherwise. The control gate controls the update of the cell, and it is defined in Equation (13). To update the hidden layer ($h_{t-1}$) and update the output, finally the output gate is used which is given by the following equations:(14)$$O_{t}=\sigma \left(w_{c}\times \left[h_{t-1},x_{t}\right]+b_{o}\right)$$
(15)$$h_{t}=O_{t}\times \tanh\left(c_{t}\right)$$

In the above equations, $x_{t}$ is the input, w represents the corresponding weight matrix of input, $b_{o}$ is the corresponding bias of input, $C_{t-1}$ is the previous block memory, $C_{t}$ represents the current block memory, $\tilde{c_{t}}$ as shown in Equation (12) is the vector of new candidate values updated with the tanh layer, $h_{t-1}$ represents the previous block output, $h_{t}$ represents the current block output. Furthermore, tanh is the hyperbolic tangent function that is used to scale the values in the range of −1 to 1, and σ is the sigmoid activation function, which gives the output between 0 and 1. We designed a sequential model containing LSTM layers with ReLU activations, a dense output layer, and an Adam optimizer with two regression loss functions. We set the input dimension in the first layer and the output dimension in the last layer of the model followed by a regression layer.

### 2.4. Loss Functions

Conventional regression loss functions are metrics-inspired losses, namely the Mean Absolute Error (MAE), and Mean Squared Error (MSE), defined as
(16)$$MAE=\frac{1}{n}\sum_{i=1}^{n}\left|p_{i}-g_{i}\right|$$
(17)$$MSE=\frac{1}{n}\sum_{i=1}^{n}{\left(p_{i}-g_{i}\right)}^{2}$$
where predicted (resp. ground truth) values are denoted $p_{i}$ (resp. $g_{i}$). We report both the loss functions, as advocated in previous publications [39,40]. The metrics to evaluate the results are Mean Absolute Error (MAE). We repeated 10-fold cross-validation with 10-folds and three repeats training the model with Adam optimizer. Models are implemented with Keras and PyTorch on NVIDIA Tesla T4 GPU.

## 3. Results and Discussion

In this section, the models discussed were evaluated on providing efficient approximations of flow velocity. Numerical experiments considered four regression models and two loss functions, in which the Mean Absolute Error (MAE) and the Mean Squared Error (MSE) were compared. Table 2 shows an average MAE value across 10 runs of experiments. Hyper-parameter optimization using grid-search with a ten-cross-validation technique was applied to optimize the number of hidden layers, hidden neurons, and the batch size. The number of hidden layers was optimized ranging from 1 to 4 and the number of hidden neurons was optimized ranging from 4 to 20 with an increment of 2. The batch sizes used in parameter tuning were 32, 64, and 128. After applying hyper-parameter optimization, the best hyper-parameters were obtained, there was one hidden layer with 10 hidden neurons, and the batch size was 32. A dropout rate of 0.02 was applied to avoid overfitting after the first hidden layer. Each of the MLP, CNN, and LSTM architectures were ran for 100, 500, 1000, and 2000 epochs. The best results were obtained with 100 epochs for MLP with an execution time of 4 min per 10 epochs and 500 epochs for both CNN and LSTM, as with a higher number of epochs, the average performance decreased.

Multiple regression algorithms were adopted to provide models capable of predicting the velocity field. The deep learning models (MLP, CNN, and LSTM) outperformed the baseline LASSO model. The hypothesis that the relationship between the 2D projection images and the initial and final velocity components can be represented as a multivariate regression problem with improved accuracy has been demonstrated in Table 2. Significant differences across the three deep learning models were not observed. All of the reported models were statistically significant with a *p*-value of< 0.01 or 0.05. Furthermore, the standard deviation of all the models was less than 1e−6 m/s. However, the MLP model was computationally less expensive and hence, a quantitative analysis of the results in comparison with the baseline OFM and with the standard CFD approach is presented in Figure 7 and Figure 8 for comparisons of both u- and v-components velocity contours. The velocity distribution (both u-component and v-component) can be used to analyze flow characteristics such as the wall shear stress, which is a key parameter in the study of the pathophysiology of vascular diseases.

As shown in Figure 7, the OFM results overestimated the u-component velocity in the central region of the domain, especially downstream, but OFM underestimated the flow between y = 0.005 m and y = 0.01 m. Despite these discrepancies in the u-contours between OFM and CFD, the predicted values were able to match the CFD values between y = 0.005 m and 0.01 m as well as further downstream. There is a discernible difference between the predicted and CFD at the dye inlet, where the predicted values overestimate near the centerline, which can be attributed by the overestimation from the OFM data. Discrepancies of the v-component velocity contours between OFM and CFD are illustrated in Figure 8, where OFM consistently underestimates the velocity especially downstream from y = 0.017 m. However, the predicted values show similar contour patterns as the CFD contour for most of the regions except near the inlet. The predicted inlet exhibits more of a “V” shape whereas the Ground contour demonstrates more of a semicircle appearance for the velocity range near −0.0075 m/s. These contour plots provide insight in the amount of error associated with projective images when analyzed with OFM as well as the accuracy improvement when ML corrections are applied to OFM results.

Furthermore, the velocity profiles at various locations of the flow field were extracted for comparison as shown in Figure 9. Near the inlet, at y = 1.66 mm, the v-component velocity profile for the MLP predictions has trouble matching the CFD results near x = ±0.7 mm to x = ±1.5 mm, where the MLP predictions have “V-shape” in the contour as discussed earlier. Further downstream at y = 6.64 mm, the predicted velocity profile resembles the ground-truth velocity profile well with little discrepancies. These quantitative comparisons of the velocity profiles can be further examined with error analysis in Figure 10 showing a significant reduction in the percentage of error with respect to the ground truth (CFD) for the v-component velocity. Error associated with the predicted values reach as high as 13%, near the inlet, and as low as 2% around y = 0.008 m. It is interesting to observe that the error in OFM is lower at the inlet and increases further downstream, whereas the opposite is true for MLP predictions. The MLP predictions presented larger errors near the inlet as a result of larger variations in the OFM velocities caused by low intensity gradients near the inlet on the image. As the dye moving downstream, the MLP predictions can match the ground truth well. The averaged V-error of about 53.5% in the velocity estimation with OFM has been significantly reduced to 2.5% in average by the current MLP predictions. The fact this model significantly reduces the error downstream of the inlet, provides high hopes for the biomedical community since the regions of high interest are further downstream, such as aneurysms and bifurcations. The current pilot study is focusing on prediction of the velocity distributions. Future studies related to the wall shear stress predictions would focus on the local velocity gradient accuracies, especially near the wall. The current ML model development focused on the 2D velocity map prediction, which limits its use for the complicated 3D flow environment, such as the blood flow in an aneurysm. Another limit is the current model can only predict the velocity, but another important flow information, pressure in the flow field, cannot be predicted. We plan to encode the Navier–Stokes equation into the modeling as recently proposed by Raissi et al. [25], as well as extend the current model to predict the 3D flow feature in the future study.

This research focused on a pilot study to examine the efficiency and accuracy of ML algorithms to predict velocity field based on projective contrast images of flows in a simplified tube model. OFM was applied at the beginning to incorporate physics-based optical flow equations into the velocity prediction, which is more efficient than encoding the Navier–Stokes equations into the ML algorithm. To apply this methodology to clinical applications, further studies with more complex geometries, such as blood vessels and aneurysms, need to be conducted. By providing additional cases and therefore training data, the authors believe this methodology can develop a ML model which could be applied directly to DSA imaging for quick high resolution velocity measurements to obtain wall shear stress measurements.

## 4. Conclusions

In this study, the flow velocity estimation as a multi-output regression problem was performed. The performance of four machine learning models on simulated CFD images as well as in vitro experimental OFM images were evaluated. Results showed that the performance of all the neural network architectures (MLP, CNN, and LSTM) is comparable. However, the MLP model is significantly less computationally intensive as compared to the deep learning counterparts. The LSTM model has the longest computation time and the best performance compared to CNN and MLP. Both MLP and LSTM models can be used to automate the flow velocity estimation depending on the task and considering the cost for computation. The pilot study in this work established a novel ML framework to predict the velocity field based on OFM velocity estimation on projective contrast images of internal flows.

The following summarizes the main contributions of this work, i.e.,
(1)We have presented an analysis that ML algorithms are able to correct OFM results from projection-based images significantly reducing the error rate.(2)We have extended the literature by considering the interaction of u- and v-components velocity with the intensity gradients on the image in both x- and y-directions.(3)We have released the data and code used in this work for reproducibility and further research in this direction.

## Figures and Tables

**Figure 1 bioengineering-09-00622-f001:**
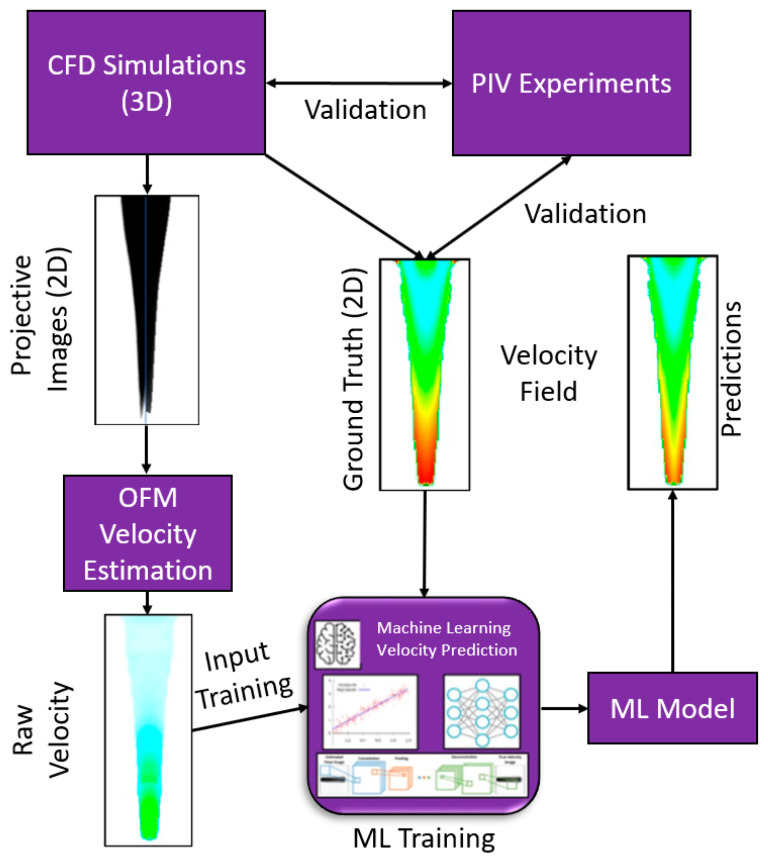
ML framework for velocity estimations on the dye perfusion.

**Figure 2 bioengineering-09-00622-f002:**
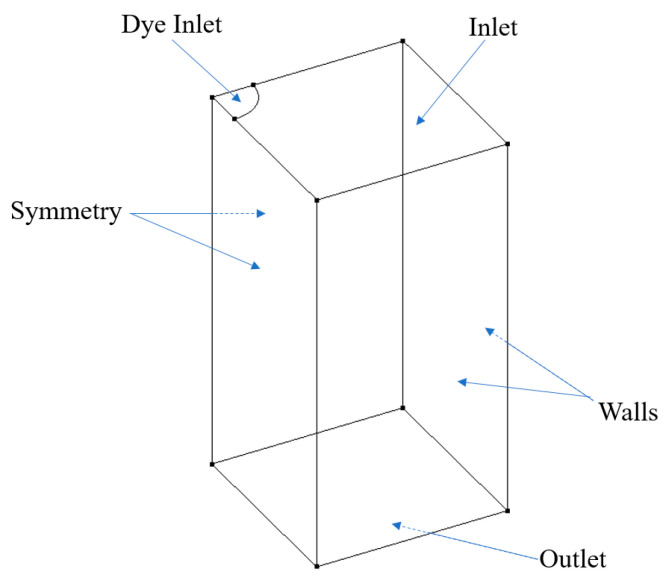
Schematic geometry for CFD simulations.

**Figure 3 bioengineering-09-00622-f003:**
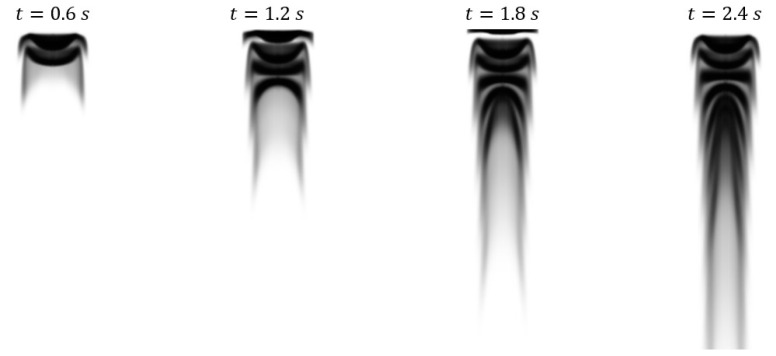
Projective images from case 4 generated from CFD results by using the Beer-Lambert law to simulate light passing through the domain.

**Figure 4 bioengineering-09-00622-f004:**
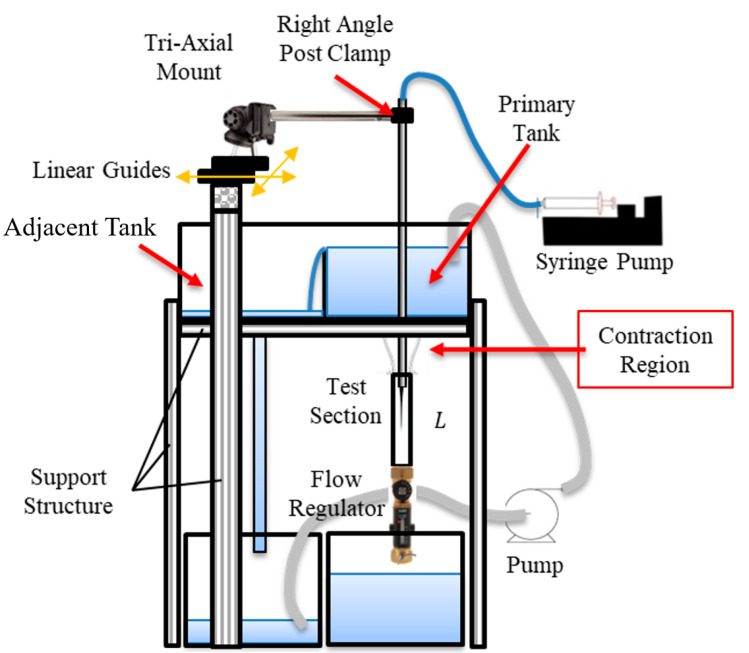
Schematic of in vitro experiments with the contraction section, dye centering apparatus, and partitioning wall to maintain constant water level.

**Figure 5 bioengineering-09-00622-f005:**
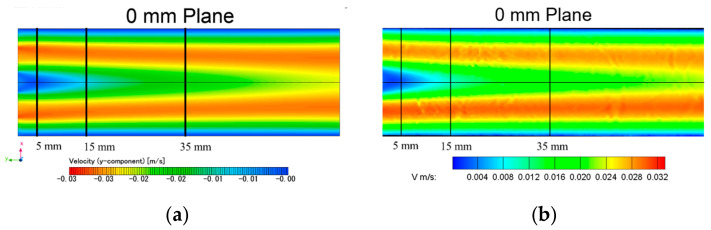
Velocity distribution comparison in the middle plane of the tube: (**a**) CFD simulation; (**b**) PIV experiment.

**Figure 6 bioengineering-09-00622-f006:**
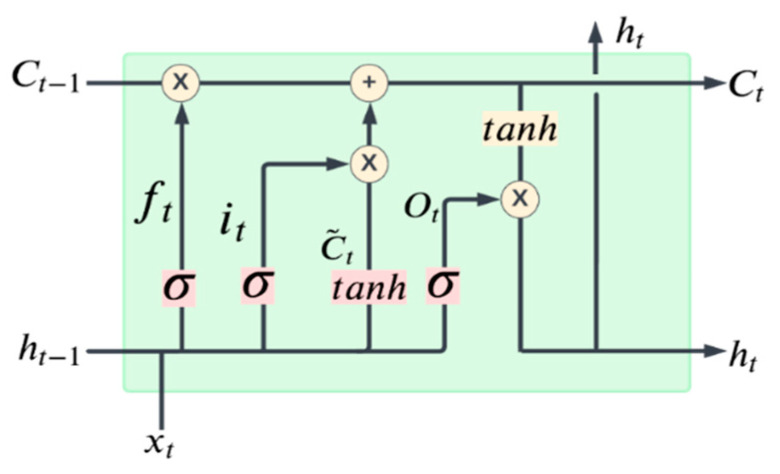
Structure of our LSTM cell at time *t*.

**Figure 7 bioengineering-09-00622-f007:**
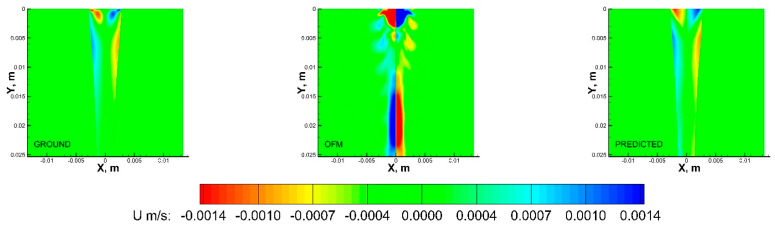
U−contours of the CFD ground truth (**left**), OFM (**middle**), and predicted (**right**) results.

**Figure 8 bioengineering-09-00622-f008:**
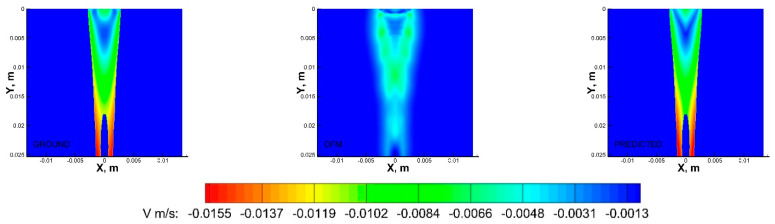
V−contours of the CFD ground truth (**left**), OFM (**middle**), and predicted (**right**) results.

**Figure 9 bioengineering-09-00622-f009:**
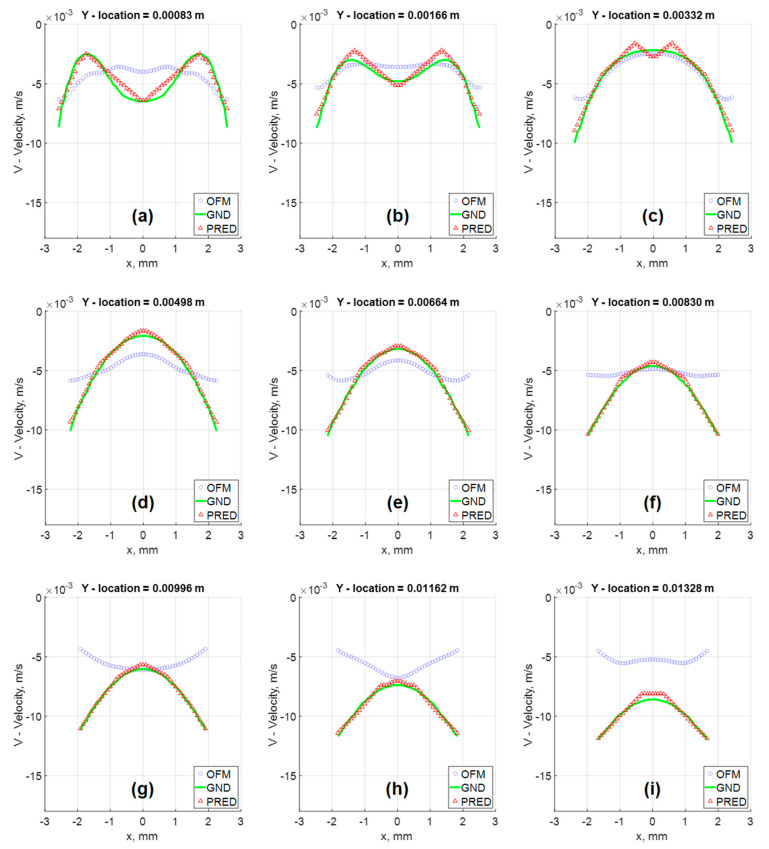
Velocity Vector fields comparison of in silico results at various Y locations: (**a**) Y = 0.00083 m; (**b**) Y = 0.00166 m; (**c**) Y = 0.00332 m; (**d**) Y = 0.00498 m; (**e**) Y = 0.00664 m; (**f**) Y = 0.00830 m; (**g**) Y = 0.00996 m; (**h**) Y = 0.01162 m; (**i**) Y = 0.01328 m.

**Figure 10 bioengineering-09-00622-f010:**
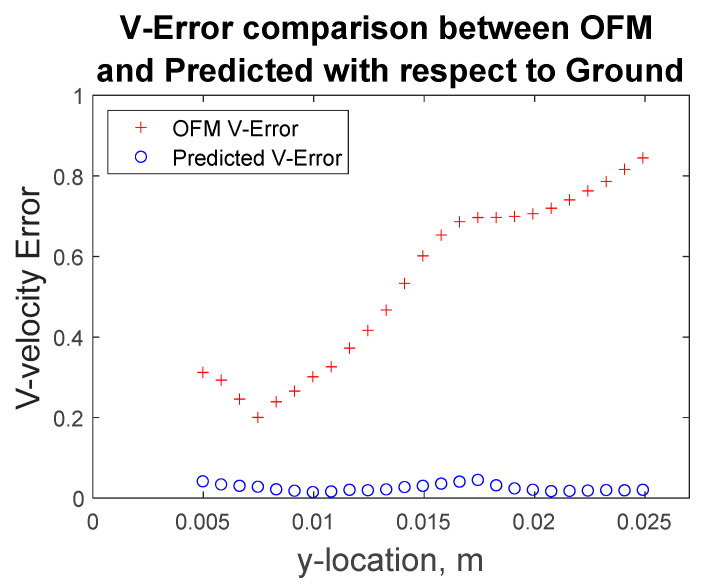
Percentage error of OFM estimation and MLP predicted results for v-component velocity with respect to the ground truth data at various Y-locations.

**Table 1 bioengineering-09-00622-t001:** V-velocity validation between CFD and PIV at 5 mm, 15 mm, and 35 mm downstream of the inlet.

Locations	5 mm	15 mm	35 mm
Relative Error (%)	3.3	4.8	3.5

**Table 2 bioengineering-09-00622-t002:** Performance of regression models in terms of mean absolute error (mae) for two different loss functions: MSE, MAE.

	LASSO	MLP	CNN	LSTM
Validation with simulated test data				
MAE (m/s)	2 × 10^−4 †^	2 × 10^−4^ *	4 × 10^−4^ *	4 × 10^−4^ *
MSE (m/s)	3 × 10^−8 †^	5 × 10^−8^ *	7 × 10^−8 †^	8 × 10^−8^ *
Validation with in vitro experimental data				
MAE (m/s)	3 × 10^−3 †^	5 × 10^−4^ *	6 × 10^−4 †^	6 × 10^−4^ *
MSE (m/s)	5 × 10^−7 †^	4 × 10^−8^ *	5 × 10^−8^ *	8 × 10^−8 †^

* indicates *p*-value < 0.01, † indicates *p*-value < 0.05.

## Data Availability

Data available on request due to restrictions, e.g., privacy or ethical.

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
