# Peer review of "Machine Learning for Aiding Blood Flow Velocity Estimation Based on Angiography"

_bioengineering, 2022, doi:10.3390/bioengineering9110622_

Round 1

Reviewer 1 Report

Thank you for giving me this opportunity to review the research article entitled, "Machine Learning for Aiding Blood Flow Velocity Estimation 2 Based on Angiography".

I here carefully reviewed the submitted set of the manuscript and found it merits for publication. However, the revisions should be necessary to meet the scientific standard for publication as raised followings.

1. Abstract should include more concrete and objective data, as summarized from this study. The whole abstract should be rewritten.

2. In the Introduction section, the background of this study seems unclear, more concrete data should be introduced and shown as background to conduct this research. To begin with, "In developed countries, the leading causes of mortality and morbidity are diseases in the vascular system", this sentence is not true!

3. Figures of apparatus and the systems recruited in this study look unclear, hard to recognize, should be revised.

4. In the discussion section, whole discussions should be rewritten with describing the relations between obtained data, results with clinical settings.

How these results can be interpretable to the clinical applications and the settings.  

Author Response

Dear reviewer 1,

Thank you for your comments concerning our manuscript titled “Machine Learning for Aiding Blood Flow Velocity Estimation Based on Angiography”. Your constructive comments and critiques are valuable and helpful for revising and improving our manuscript. Updates are highlighted in the revised manuscript. Responses to your comments are as follows specifically:

Comments in summary: I here carefully reviewed the submitted set of the manuscript and found it merits for publication. However, the revisions should be necessary to meet the scientific standard for publication as raised followings.

Response: Thanks very much for your suggestions. Accordingly, we made necessary revisions which have been highlighted in the updated manuscript.

Comment #1: Abstract should include more concrete and objective data, as summarized from this study. The whole abstract should be rewritten.

Response: We agreed with the reviewer’s opinion, and the revised Abstract section has been highlighted in the updated manuscript.

Comment #2: In the Introduction section, the background of this study seems unclear, more concrete data should be introduced and shown as background to conduct this research. To begin with, "In developed countries, the leading causes of mortality and morbidity are diseases in the vascular system", this sentence is not true!

Response: We understood the reviewer’s concern. We checked and improved the Introduction section thoroughly, and updates have been highlighted in the revised manuscript.

Comment #3: Figures of apparatus and the systems recruited in this study look unclear, hard to recognize, should be revised.

Response: To clear the reviewer’s concern, we have revised Figures 1 and 4  to better show apparatus and the framework of this study. In addition, we added Figure 2 to show the schematic geometry for CFD simulations. Updates have been highlighted in the revised manuscript.

Comment #4: . In the discussion section, whole discussions should be rewritten with describing the relations between obtained data, results with clinical settings.

Response: We understood the reviewer’s concern. While as a pilot study, this study aimed to build a ML framework to test the possibility of using AI to predict hemodynamic information and then to assist clinical diagnosis. While the framework still needs to be improved and the current database was not from the patient-specific clinical images. Too many discussions with the clinical settings may be not appropriate in the current study. While our further investigations will pay close attention to the clinical applications using the proposed ML algorithms. To address the reviewer’s concern, we made necessary revisions which were highlighted in the revised manuscript.

Comment #5: How these results can be interpretable to the clinical applications and the settings.  

Response: This research focused on a pilot study to examine the efficiency and accuracy of ML algorithms to predict velocity fields based on projective contrast images, which are similar as preclinical angiographic images (i.e., CTA/MRA/DSA). Once the proposed ML algorithms can be validated and improved by more data sets using realistic vascular models and patient-specific blood flow conditions, this methodology can be used to aid clinical diagnosis by quantify risks induced by hemodynamic factors in the cardiovascular system in a noninvasive, efficient and accurate manner. We hope our explanation can address the reviewer’s doubts.

Reviewer 2 Report

In this work, the dye injection in a water flow was simulated as an analogy of the contrast perfusion in blood flow using CFD. The simulations provided velocity field and the flow images with dye patterns. A rough velocity field was estimated using OFM based on projective images. ML algorithams are trained using the CFD data and OFM velocity estimation as the input.

I've had a very pleasent task to review this paper, which I find very interesting. This is an extensive research, with a lot of numerical analysis. 
Thematically the work is interesting for the researchers and professionals and the proposed manuscript is relevant to the scope of the journal.

I found it appropriate for publication in the Bioengineering journal, but only after some modifications and clarification from the Authors.
The title is a clear representation of the manuscript's content. The abstract reflects realistically the substance of the work. 
The list of keywords could be improved, by adding (or changing) one or two more terms.

The overall organization and structure of the manuscript are appropriate. The paper is well written and the topic is appropriate for the journal.
The aim of the paper is well described and the discussion was well approached, its results and discussion are correlated to the cited literature data.

A few newer literature references could be added and discussed.

The novelty of the work must be more clearly demonstrated.

The significance of the Work: Given the large number of analyzed data, this is an interesting study with a possible significant impact in this area.

Statistical interpretation of the analytical data must be more properly presented. The verification of the model should be performed. Model validation is possibly the most important step in the model building sequence.

Other Specific Comments: The work is properly presented in terms of the language. The work presented here is very interesting and well done, it is presented in a compact manner.

In general, there are no doubtful or controversial arguments in the manuscript. The methodology applied in the research is presented in clear manner, so that it is repeatable by other authors.
The results are presented in a logical sequence and the discussion and analysis of the results are properly elaborated. The claims in the section "Conclusion" are reasonable and supported by the presented results.
The main drawback of the paper i s the extent of novelty, or the main novelty in the present work, compared to the works of other researchers? In my opinion, the authors should put additional effort to demonstrate that the present work gives a substantial contribution in the research area.

Author Response

Dear reviewer 2,

Thanks for your comprehensive reviewing our manuscript titled “Machine Learning for Aiding Blood Flow Velocity Estimation Based on Angiography”. We are encouraged by the positive comments such as “This is extensive research, with a lot of numerical analysis. Thematically the work is interesting for the researchers and professionals and the proposed manuscript is relevant to the scope of the journal”. Also, we appreciate the reviewer with other comments and critiques which are very helpful for improving our manuscript. Updates are highlighted in the revised manuscript. The responses to your comments are as follows:

Overall comment: In this work, the dye injection in a water flow was simulated as an analogy of the contrast perfusion in blood flow using CFD. The simulations provided velocity field and the flow images with dye patterns. A rough velocity field was estimated using OFM based on projective images. ML algorithms are trained using the CFD data and OFM velocity estimation as the input. I've had a very pleasant task to review this paper, which I find very interesting. This is extensive research, with a lot of numerical analysis. Thematically the work is interesting for the researchers and professionals and the proposed manuscript is relevant to the scope of the journal. I found it appropriate for publication in the Bioengineering journal, but only after some modifications and clarification from the Authors. The title is a clear representation of the manuscript's content. The abstract reflects realistically the substance of the work.

Response: Thanks very much for the reviewer agreeing with the research work we performed, which encourages us to carry out future research work greatly.

Comment #1: The list of keywords could be improved, by adding (or changing) one or two more terms.

Response: We agreed with the reviewer’s suggestion. We add three more keywords: least absolute shrinkage and selection operator (LASSO), angiography, particle image velocimetry (PIV), and the updates are highlighted in the revised manuscript.

Comment #2: The overall organization and structure of the manuscript are appropriate. The paper is well written, and the topic is appropriate for the journal. The aim of the paper is well described, and the discussion was well approached, its results and discussion are correlated to the cited literature data. A few newer literature references could be added and discussed.

Response: To address the reviewer’s concerns, we added more literatures which were published recently, especially for recent applications of machine learning (ML) algorithms in the predictions of vascular diseases. The updates have been highlighted in Introduction section of the revised manuscript.

Comment #3: The novelty of the work must be more clearly demonstrated.

Response: To address the reviewer’s concern, the novelty has been clarified in sections of Abstract, Introduction and Conclusion in the revised manuscript. Specifically, as a pilot study, this work built a framework of ML algorithms to predict the velocity field in a simplified in-vitro tube model using projective images, and the PIV validated CFD results were served as the ground truth data. The results showed that the employed ML model can estimate the v-velocity (main component) accurately by the reducing relative error 53.5% to 2.5% in average. The ML framework have high potential to provide an alternative pathway to aid clinical diagnosis by predicting hemodynamic information with features in high efficiency and accuracy.

Comment #4: The significance of the Work: Given the large number of analyzed data, this is an interesting study with a possible significant impact in this area. Statistical interpretation of the analytical data must be more properly presented. The verification of the model should be performed. Model validation is possibly the most important step in the model building sequence.

Response: We agreed with the reviewer’s perspective that validation is the most important in the model building processing.  To address the reviewer’s concern, the error analysis for CFD model validation was conducted and shown in Figure 5 and Table 1. In addition, Figure 10 shows the error comparisons in velocity estimations among the CFD, OFM, and ML predicted were investigated statistically. All updates were highlighted in the revised manuscript.

Other Specific Comments: The work is properly presented in terms of the language. The work presented here is very interesting and well done, it is presented in a compact manner. In general, there are no doubtful or controversial arguments in the manuscript. The methodology applied in the research is presented in clear manner, so that it is repeatable by other authors.
The results are presented in a logical sequence and the discussion and analysis of the results are properly elaborated. The claims in the section "Conclusion" are reasonable and supported by the presented results. The main drawback of the paper is the extent of novelty, or the main novelty in the present work, compared to the works of other researchers? In my opinion, the authors should put additional effort to demonstrate that the present work gives a substantial contribution in the research area.

Response: Thanks very much for the reviewer’s suggestion. Similar to the response to Comment #3, the novelty has been clarified in sections of Abstract and Introduction in the revised manuscript. This research focused on a pilot study to examine the efficiency and accuracy of ML algorithms to predict velocity fields based on projective contrast images which are similar for preclinical angiographic images (i.e., CTA/MRA/DSA). Once the proposed ML algorithms can be validated and improved by more data sets using realistic vascular models and patient-specific blood flow conditions, this methodology can be used to aid clinical diagnosis by quantify risks induced by hemodynamic factors in the cardiovascular system in a noninvasive, efficient, and accurate manner. These investigations have not been done well previously and still need more efforts from related research communities continuously. We hope our updates and explanation can clear the reviewer’s concerns.
